

# Construction of a ceRNA network of regulated ferroptosis in doxorubicin-induced myocardial injury

Hongwei Ye[1,2], Yuping Li[1,2], Lu Li[1,2], Yuhui Huang[1,2], Jiahui Wang[2,3] and Qin Gao[1,2]

[1] Department of Physiology, Bengbu Medical College, Bengbu, Anhui, China
[2] Key Laboratory of Basic and Clinical Cardiovascular Diseases, Bengbu Medical College, Bengbu, Anhui, China
[3] Department of Anatomy, Bengbu Medical College, Bengbu, Anhui, China

Corresponding author
Qin Gao, bbmcgq@126.com

## ABSTRACT

**Background:** Ferroptosis and long-noncoding RNAs (lncRNAs) play crucial roles in doxorubicin (DOX)-induced myocardial injury (DIMI). Nevertheless, there is no research to construct competing endogenous RNAs (ceRNAs) network between lncRNAs and ferroptosis-related key gene. So our research was designed to screen ferroptosis-related genes from differentially expressed mRNAs in DIMI and construct lncRNAs regulated ferroptosis-related key gene ceRNAs network.

**Methods:** The male mice were injected with DOX intraperitoneally to induce myocardial injury, myocardial injury was evaluated by hematoxylin and eosin (HE) staining, and ferroptosis-related protein-glutathione peroxidase 4 (GPx4) protein expression was detected. The differentially expressed lncRNAs and mRNAs were detected by microarray, and the ferroptosis-related genes were screened to construct a protein-protein associations (PPA) network, the highest maximal clique centrality (MCC) score gene were identified by Cytoscape software, miRNAs bound to key genes and lncRNAs bound to miRNAs were predicted; then, the obtained lncRNAs were intersected with differentially expressed lncRNAs detected by microarray. Finally, the lncRNA/miRNA/mRNA ceRNA network of the highest MCC score gene regulating ferroptosis in DIMI was constructed. The expressions of the key components in ceRNA network were detected by qRT-PCR.

**Results:** Compared with the control group, in the DOX group, myocardial enzymes and HE staining showed that myocardium structure was changed, and GPx4 protein expression was decreased. The differentially expressed 10,265 lncRNAs and 6,610 mRNAs in the DOX group were detected *via* microarray. Among them, 114 ferroptosis-related genes were obtained to construct PPA networks, and Becn1 was identified as the key gene. Finally, the ceRNA network including Becn1, three miRNAs and four lncRNAs was constructed by predicting data of the Starbase database. The relative expressions of these components in ceRNA net were up-regulated and consistent with microarray results.

**Conclusions:** Based on the microarray detection results and bioinformatics analysis, we screened ferroptosis-related gene Becn1 and constructed the lncRNA/miRNA/mRNA ceRNA network of regulated ferroptosis in DIMI.

# INTRODUCTION

Doxorubicin (DOX) is a classical first-line anti-tumor drug, and is widely used in clinical to treat acute leukemia, lung cancer, breast cancer, bladder cancer, gastric cancer, liver cancer and other tumors due to its high efficiency and wide spectrum (*Zhu & Lin, 2021*). However, the dose-dependent cardiotoxicity limits the clinical application of DOX. In view of the universality and importance of DOX in clinical antitumor therapy, it is of great significance to explore the potential mechanisms of DOX-induced myocardial injury (DIMI) and seek for the effective measures to prevent the happening of cardiotoxicity.

Ferroptosis as a novel form of cell death has attracted widespread attention in recent years. It is characterized by excessive accumulation of intracellular lipid reactive oxygen species (ROS) and lipid peroxidation induced by glutathione peroxidase4 (GPx4) inactivation. Studies have shown that ferroptosis is closely related to the occurrence and development of Alzheimer's disease, tumor and stroke (*Liu et al., 2020*). Ferroptosis plays an important role in cardiovascular diseases, such as resveratrol protects against myocardial I/R injury *via* reducing oxidative stress and attenuating ferroptosis (*Li et al., 2022*), ferritinophagy-mediated ferroptosis is involved in the development of sepsis-induced cardiac injury (*Li et al., 2020*). Ferroptosis also plays a crucial role in DIMI. It has been reported that mitochondrial-dependent ferroptosis plays a key role in the progression of DIMI (*Tadokoro et al., 2020*), epigallocatechin gallate pretreatment alleviates DOX-induced ferroptosis and cardiotoxicity by upregulating AMPKα2 and activating adaptive autophagy (*He et al., 2021*), knockout of TRIM21 can reduce DOX cardiotoxicity by inhibiting ferroptosis (*Hou et al., 2021*). However, the signaling pathways and the pathogenesis of DOX-mediated ferroptosis and cardiac failure remain largely unknown.

Long-noncoding RNAs (lncRNAs) are a subset of non-coding RNAs, the more and more emerging evidences have suggested that lncRNAs could serve as sponges for miRNAs through miRNA response elements, resulting in alterations in miRNAs-regulated mRNA levels, and the lncRNA/miRNA/mRNA competing endogenous RNAs (ceRNA) network is reported to be one of the important mechanisms in the development and progression of cardiovascular diseases. The ceRNA regulatory mechanism is also involved in DIMI. *Xia et al. (2020)* found that lncRNA-MALAT1/miR-92A-3p/ATG4a partially mediated the protective effect of exosomes secreted by hypoxic pretreated mesenchymal stem cells on DIMI. Therefore, the enrichment and discovery of lncRNA/miRNA/mRNA ceRNA networks may help to reveal the potential function of lncRNAs involved in DIMI.

In this study, we detect the differentially expressed lncRNAs and mRNAs in DIMI by microarray analysis, and then screen ferroptosis-related genes through bioinformatics analysis; we aim to construct the ceRNA network to explore the potential mechanism, and hope to provide new clues for studies on the role of lncRNAs in regulating ferroptosis in DIMI.
## MATERIALS AND METHODS

### Animals

Male C57BL/6J mice (body weight of 18–22 g) were purchased from Henan Skbex Biotechnology Co., LTD, Zhengzhou, Henan, China. All animals were maintained in the SPF animal laboratory, and housed using standard cages in the environment of the standard humidity/temperature and a 12–12h light-dark cycle and fed free access to sterile rodent food and water. After acclimatization to the environment for one week, the mice were used for the experiment. All animal experiments were approved by the Animal Management and Ethics Committee of Bengbu Medical College (Permit number: [2022] 024), and the care and treatment of the animals were carried out in strict accordance with the Regulations on the Management of Experimental Animals.

The twelve mice were randomized into two groups: control group (CON) and DOX group (DOX), each consisted of six mice. The mice were given DOX (15 mg/kg (*Hu et al., 2019*), purchased from Dalian Meilun Biotechnology Co., LTD, Dalian, China.) in the DOX group, and given the same dose saline in the CON group through single intraperitoneal injection. After 3 days of intraperitoneal injection of DOX or saline, blood was collected through eye vessels by removing eyeballs when the mice were anesthetized with 1.5% isoflurane *via* a mask. Then, the animals were sacrificed by cervical dislocation, and heart tissues were excised for further detection. The overall experimental design and experimental workflow were shown in Fig. 1.

### Serum lactic dehydrogenase (LDH) and myocardial-bound creatine kinase (CK-MB) levels detection

After 3 days of intraperitoneal injection of DOX or saline, the mice were anesthetized with isoflurane to collect blood through eye vessels by removing eyeballs. Blood was left at room temperature for 2 h, then serum was obtained by centrifugating the blood at 1,000 g for 15 min at 4 °C after coagulation. Serum LDH and CK-MB levels, which reflect the damage degree of myocardial injury, were determined according to the kit instructions (purchased from Nanjing Jiancheng Bioengineering Institute, Nanjing, China) in the two groups.

### Histopathology observation through hematoxylin and eosin (HE) staining method

The hearts were removed immediately after anesthesia and cleaned with phosphate buffered saline (PBS) at 4 °C, the left ventricular myocardium was selected and fixed with 4% paraformaldehyde for 48 h, dehydrated by gradient ethanol, embedded in paraffin, sliced (0.5 mm) and stained with hematoxylin and eosin. The histopathologic damages of myocardial tissue were observed under light microscope (Nikon Eclipse E100). All sections were assessed for the presence of myocardial injury in a blinded fashion.

### GPx4 protein expression measurement by western blot analysis

Mouse myocardial tissue (40 mg) was homogenized in ice-cold RIPA lysate (500 µl) containing phenylmethylsulfonyl fluoride (PMSF), and centrifuged at 12,000 g for 5 min to collect supernatant. The concentration of total protein was determined by bicinchonininc

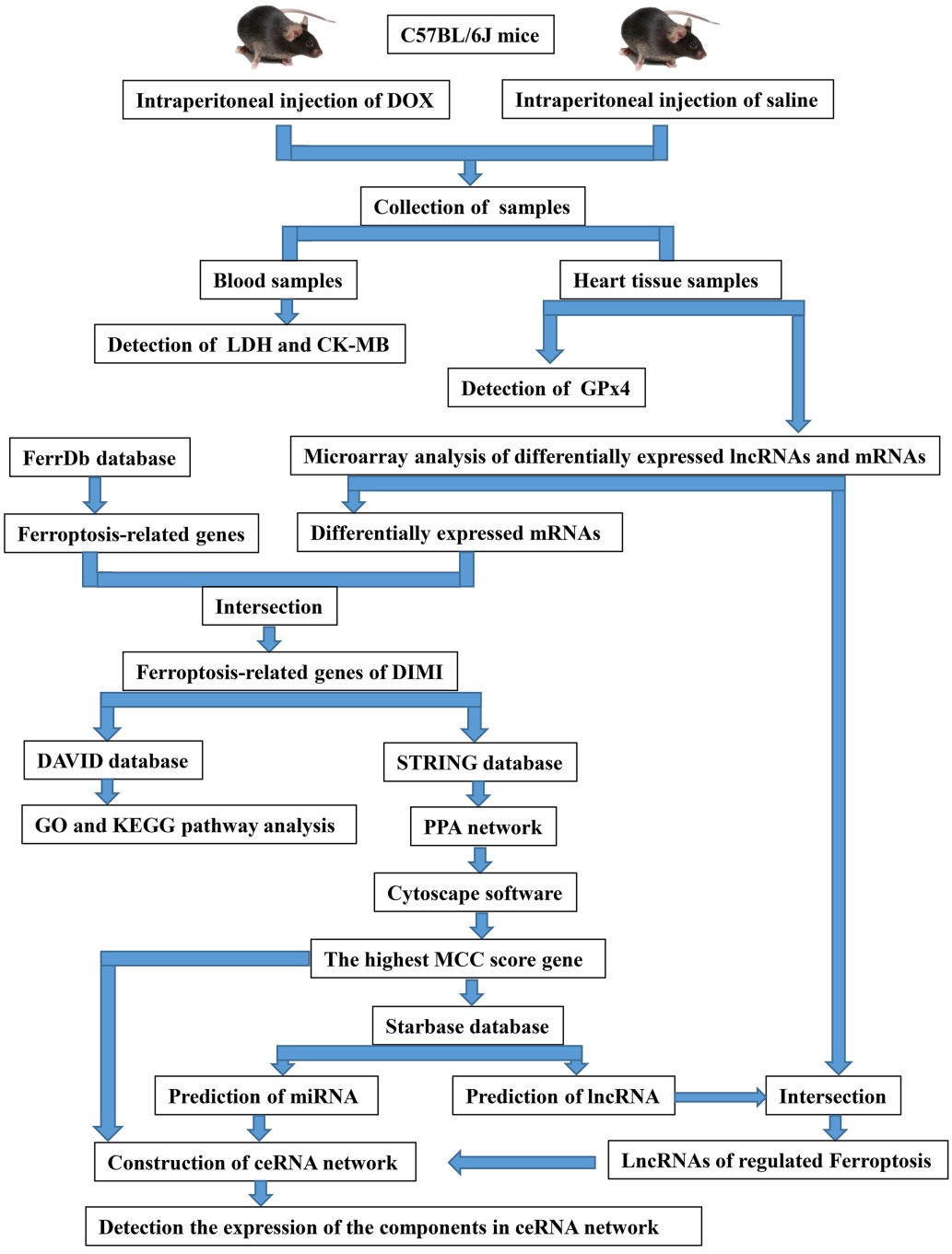

**Figure 1 The overall experimental design and experimental workflow.** LDH, lactic dehydrogenase; CK-MB, myocardial-bound creatine kinase; GPx4, glutathione peroxidase 4; DIMI, DOX-induced myocardial injury; PPA network, protein-protein associations network; MCC, maximal clique centrality; ceRNA, competing endogenous RNA.

acid (BCA) kit, the obtained total protein was added into western blot loading buffer, and boiled for 5 min at 95 °C, the total protein in each groups were separated by SDS-PAGE with 10% polyacrylamide gel for 2 h, then electro-transferred to polyvinylidene fluoride (PVDF) membranes, and blocked by 5% skim milk, and incubated with primary antibody

of anti-GPx4 (1:3,000; Abcam) and anti-glyceraldehyde 3-phosphate dehydrogenase (GAPDH) (1:5,000; Absin) at 4 °C overnight, then the membranes were incubated with second antibody (1:8,000; Absin) at room temperature for 1 h, and the membranes were washed by Tris-Buffered Saline with Tween 20 (TBST) for four times, finally, the bounds were stained with enhanced chemiluminescence (ECL) reagent, and was visualized using the BIO-RAD ChemiDoc Touch Imaging System (BIO-RAD, USA). The band intensity was analyzed by ImageJ software (version 1.51, NIH, USA), GAPDH was used as an internal reference for protein standardization, the relative protein expression level was calculated by the ratio of GPx4/GAPDH.

## Detection of differentially expressed lncRNAs and mRNAs by microarray

Total RNA of myocardial tissue in each group was isolated using TRIzol reagent (Invitrogen, Grand Island, NY, USA), RNA isolation methods had been described previously by our laboratory (*Hu et al., 2017*). The differentially expressed lncRNAs and mRNAs were analyzed by Kangchen Biotech Co., Ltd. (Shanghai, China) using Mouse lncRNA microarray V4.0. Briefly, each RNA sample was transcribed into fluorescent cRNA, and was labeled, hybridized to the lncRNA expression microarray, microarray images were analyzed using Agilent Feature Extraction software. The differentially expressed mRNAs and lncRNAs in the DOX group were defined according to the thresholds of an absolute fold change. The false discovery rate (FDR) controlling was a practical and powerful approach in multiple testing, which could effectively control the false positive rate. The threshold values we used to define up-regulation or down-regulation were fold change >2 and FDR < 0.05.

## Screening and biological function analysis of ferroptosis-related genes

Ferroptosis-related genes, including driver, suppressor and marker, were obtained from FerrDb database (Version 2) (http://www.zhounan.org/ferrdb/), which were intersected with the differentially expressed mRNAs in DOX group to screen the ferroptosis-related genes. These genes were screened out and analyzed using the DAVID database (Version 2021) (https://david.ncifcrf.gov/summary.jsp) to obtain its biological function in Gene Ontology (GO) functional annotations and Encyclopedia of Genes and Genomes (KEGG) signal pathway. FDR < 0.05 indicated that the GO analysis or KEGG pathway analysis were significantly enriched.

## Construction of protein-protein associations (PPA) network and screening the highest maximal clique centrality (MCC) score gene

The PPA network was constructed using the STRING database (https://cn.string-db.org/) (Version 11.5) for genes involved in regulating ferroptosis in DIMI. The data of PPA network were imported into Cytoscape software (version 3.9.1) to visualize PPA network, then the score of ferroptosis-related genes in PPA network were calculated using the CytoHubba plugin, and the highest MCC score gene was screened out.

**Table 1 Primer sequences of components in ceRNA network and GAPDH.**

| Gene name | Category | Prime sequence | Product length (bp) |
|---|---|---|---|
| Becn1 | mRNA | F:5′ GGTCCTGGGCGGAAGTCTT3′<br>R:5′ CTTAGACCCCTCCATGCCTCA3′ | 166 |
| Rian | lncRNA | F:5′ GTCCCACAGAGCATCACTATCA3′<br>R:5′ TGTCTGTATCGTCCCTCCTTCT3′ | 241 |
| Tug1 | lncRNA | F:5′AGTGAACTACGGTACTTGCCAT3′<br>R:5′CCAGGTGAAGAATCACAGAAGT3′ | 105 |
| Malat | lncRNA | F:5′GATTGTAAAGGGAGGTTTTGTGA3′<br>R:5′TCTCCAAATACTAGCCTAACCTCA3′ | 159 |
| H19 | lncRNA | F:5′CCCACCTCATTTGTCTTTATTC3′<br>R:5′TGAGTCTGCTCTTTCAAAATGTT3′ | 80 |
| GAPDH | | F:5′CACTGAGCAAGAGAGGCCCTAT3′<br>R:5′GCAGCGAACTTTATTGATGGTATT3′ | 144 |

**Note:**
Abbreviations: F, Forward; R, Reverse.

## Construction of lncRNA/miRNA/mRNA ceRNA network

The miRNAs binding with the highest MCC score ferroptosis-related gene were predicted using the Starbase database (Version 3) (https://starbase.sysu.edu.cn/index.php), and the lncRNAs binding with the predicted miRNA were predicted by Starbase database (Version 3), and the predicted lncRNA was intersected with the differentially expressed lncRNAs, finally lncRNA/miRNA/mRNA ceRNA network was obtained, and visualized using Cytoscape software (version 3.9.1; https://cytoscape.org/).

## Detection the expressions of the components in ceRNA network by qRT-PCR

The expressions of the components in ceRNA network were detected by qRT-PCR. Each component of ceRNA was reversely transcribed to cDNA using SuperScriptTM III Reverse Transcriptase (Invitrogen). The expression of lncRNAs and mRNAs was measured by qRT-PCR using a SYBR Green QPCR Supermix (Bio-Rad) on a ViiA 7 Real-time PCR System (Applied Biosystems) following the protocol: denaturation (10 min, 95 °C), and 40 amplification cycles (10 s at 95 °C, and 60 s at 60 °C). The result of each sample was normalized by GAPDH and the data were calculated by $2^{-\Delta\Delta Ct}$ method. The primers of the key components in ceRNA network and GAPDH were designed and synthesized by GenechemBio (Shanghai, China), and listed in Table 1.

## Statistical analysis

All research data were shown as mean ± SD. Independent Student's t test was used to analyze the difference between CON and DOX groups. The statistical analysis was carried out with GraphPad Prism software 8.0 (GraphPad Software Inc., San Diego, CA, USA). $P < 0.05$ was considered statistically significant.

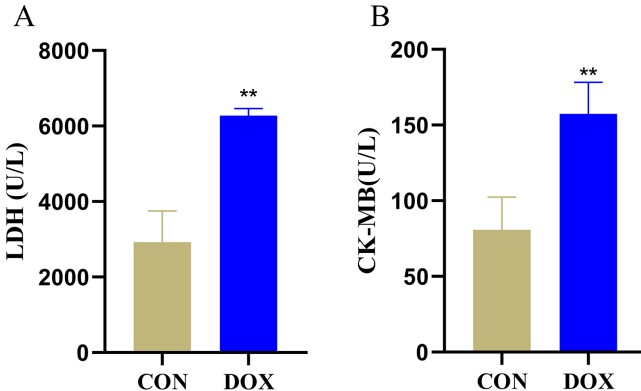

**Figure 2 Serum lactic dehydrogenase (LDH) and myocardial-bound creatine kinase (CK-MB) levels in each group.** (A) Serum LDH levels. (B) Serum CK-MB levels. $^{**}P < 0.01$ *vs.* the CON group, the values are presented as mean ± SD ($n$ = 6). CON: Con group, marked with yellow bar; DOX: DOX group, marked with bule bar.

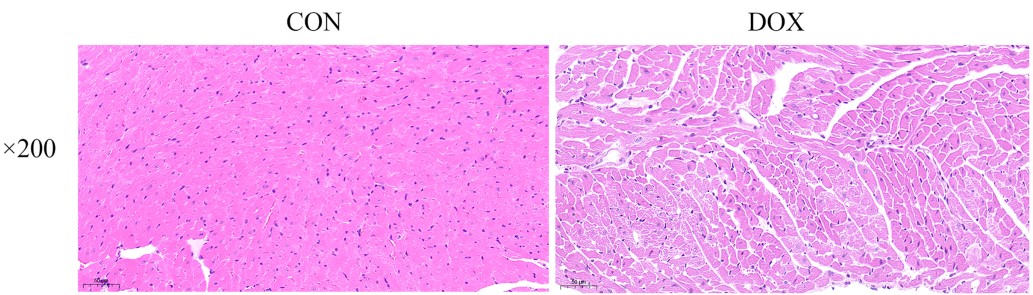

**Figure 3 Typical HE staining pictures of mice myocardial tissue in each group (×200).** CON, Con group; DOX, DOX group.

## RESULTS

### Changes of serum LDH and CK-MB levels

LDH and CK-MB are the key myocardial enzymes which existed in cytoplasm in normal condition, they could be released into the blood during myocardial injury (*Fang et al., 2019a*). In order to evaluate DIMI, we detected serum LDH and CK-MB levels. In comparison with the CON group, the levels of serum LDH and CK-MB were significantly increased in the DOX group (Fig. 2), these results indicated that DOX could induce myocardial injury.

### Changes of myocardial histological observation by HE staining

The HE staining results showed in the CON group, the myocardial tissue was uniformly stained with red staining in cytoplasm, blue staining in the nucleus, complete cell membrane, compact arrangement, and no interstitial edema. Compared with the CON group, in the DOX group, the myocardial tissue was not uniformly stained, with weakly staining cytoplasm and nucleus staining in damaged areas, and the cells were fragmented with incomplete cell membrane, and myocardial fibers were loosely arranged, interstitial edema was observed (Fig. 3). These results suggested that DOX caused myocardial structural changes.

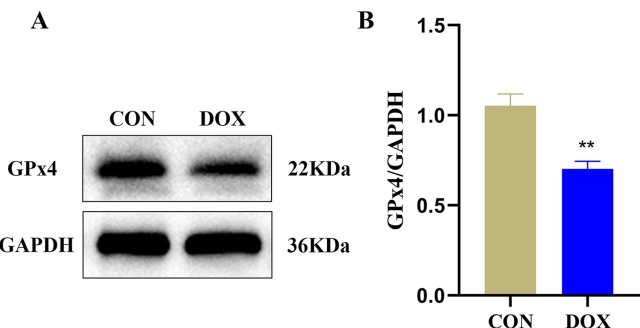

**Figure 4 The glutathione peroxidase 4 (GPx4) protein expression in mouse myocardium.**
(A) Representative western blot band. (B) The relative protein expression levels of GPx4. **$P < 0.01$
*vs*. the CON group, the values are presented as mean ± SD ($n = 3$). CON: Con group, marked with yellow
bar; DOX: DOX group, marked with bule bar.     

## Changes of GPx4 protein expression in heart tissue measured by western blot

GPx4 is an endogenous scavenger for lipid peroxides, it plays a key role in regulating
ferroptosis (*Tadokoro et al., 2020*). We measured GPx4 protein expression in the two
groups by western blot. The result showed compared with the CON group, the expression
of GPx4 protein, a marker of ferroptosis, was decreased in the DOX group (Fig. 4), it
suggested that DOX induced the happening of ferroptosis.

## Detection of differentially lncRNAs and mRNAs by microarray analysis in heart tissue

Seven myocardial samples (three in CON group and four in DOX group) were collected for
microarray analysis. The differentially expressed lncRNAs and mRNAs were shown using
cluster heatmaps and volcano plots (Fig. 5) in the DOX and CON groups. Compared with
the CON group, the differential expression of lncRNAs including 5,953 up-regulated and
4,312 down-regulated lncRNAs, and the differentially expressed 2,136 up-regulated and
4,474 down-regulated mRNAs with more than two folds change and FDR less than 0.05
were identified in the DOX group.

## Ferroptosis-related genes were screened by FerrDb database and biological functions were analyzed through DAVID database

There were 388 ferroptosis-related genes obtained by FerrDb database (Version 2; http://
www.zhounan.org/ferrdb/), which were intersected with 6,610 differentially expressed
mRNAs obtained by microarray analysis, and 114 ferroptosis-related genes in the DOX
group were obtained (Fig. 6A). Gene Ontology (GO) analysis, including biological
processes, molecular functions and cellular components, and Kyoto Encyclopedia of Genes
and Genomes (KEGG) pathways of these 114 genes were performed to reveal the potential
biological function through DAVID database (Version 2021). GO analysis results showed
that 32 biological process (BP) items, 15 cellular Component (CC) items and 11 molecular
function (MF) items were enriched. The top 10 GO enrichment items from each category
(BP, CC, MF) were shown in a bar graph in Fig. 6B, where the autophagy (GO:0006914),

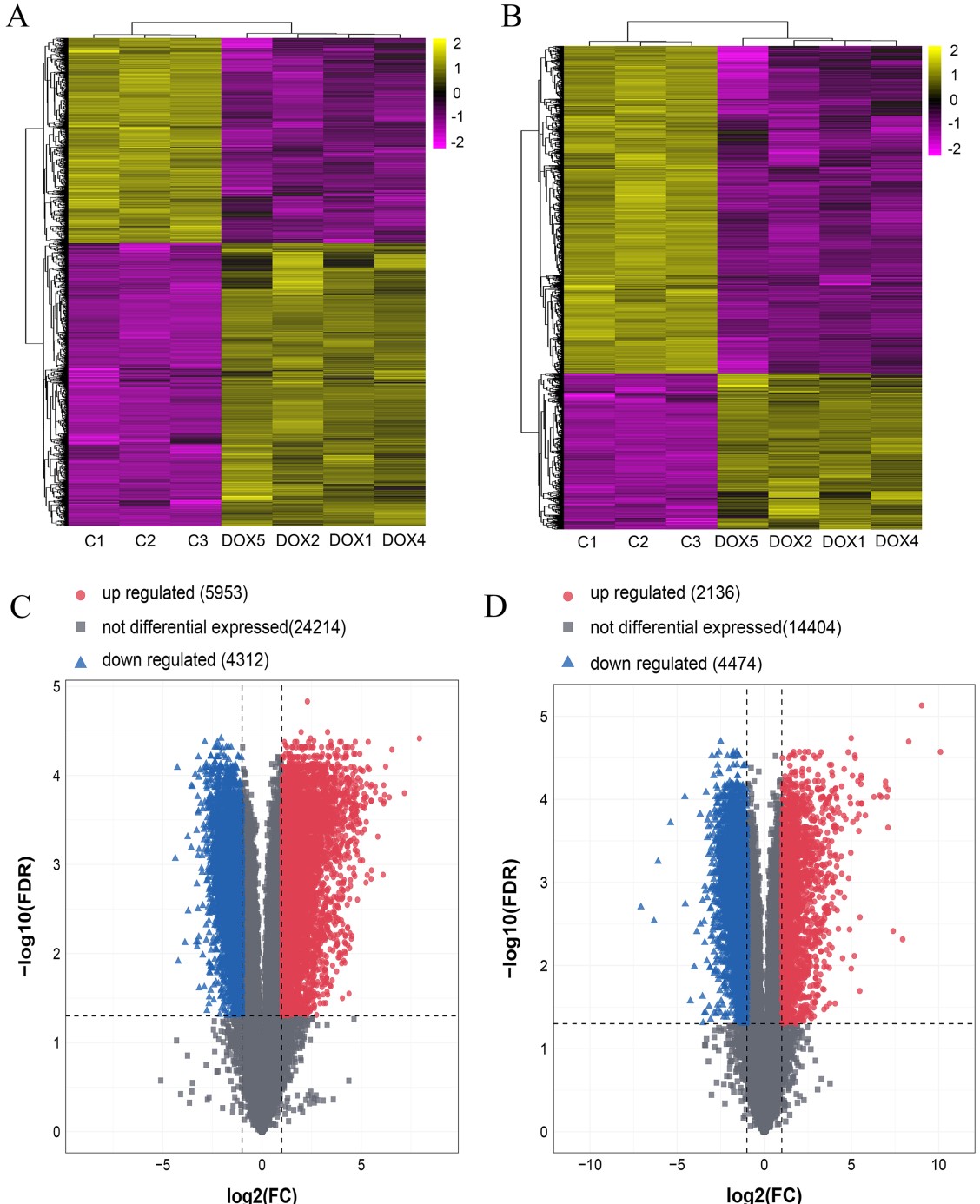

**Figure 5  The expression profiles of lncRNAs and mRNAs in mouse myocardium between CON and DOX groups.** (A) Hierarchical clustering of the differentially expressed lncRNAs. (B) Hierarchical clustering of the differentially expressed mRNAs. (C) Volcano plots of differentially expressed lncRNAs. (D) Volcano plots of differentially expressed mRNA. Hierarchical clustering and volcano plots were created by R software (*R Core Team, 2021*); up-regulated lncRNAs and mRNAs were highlighted in yellow and down-regulated lncRNAs and mRNAs were highlighted in purple in hierarchical clustering. up-regulated lncRNAs and mRNAs were marked with red dots, down-regulated lncRNAs and mRNAs were marked with blue triangle in volcano plots. The differentially expressed lncRNAs and mRNAs were selected with thresholds of fold change >2.0 and false discovery rate (FDR) < 0.05.
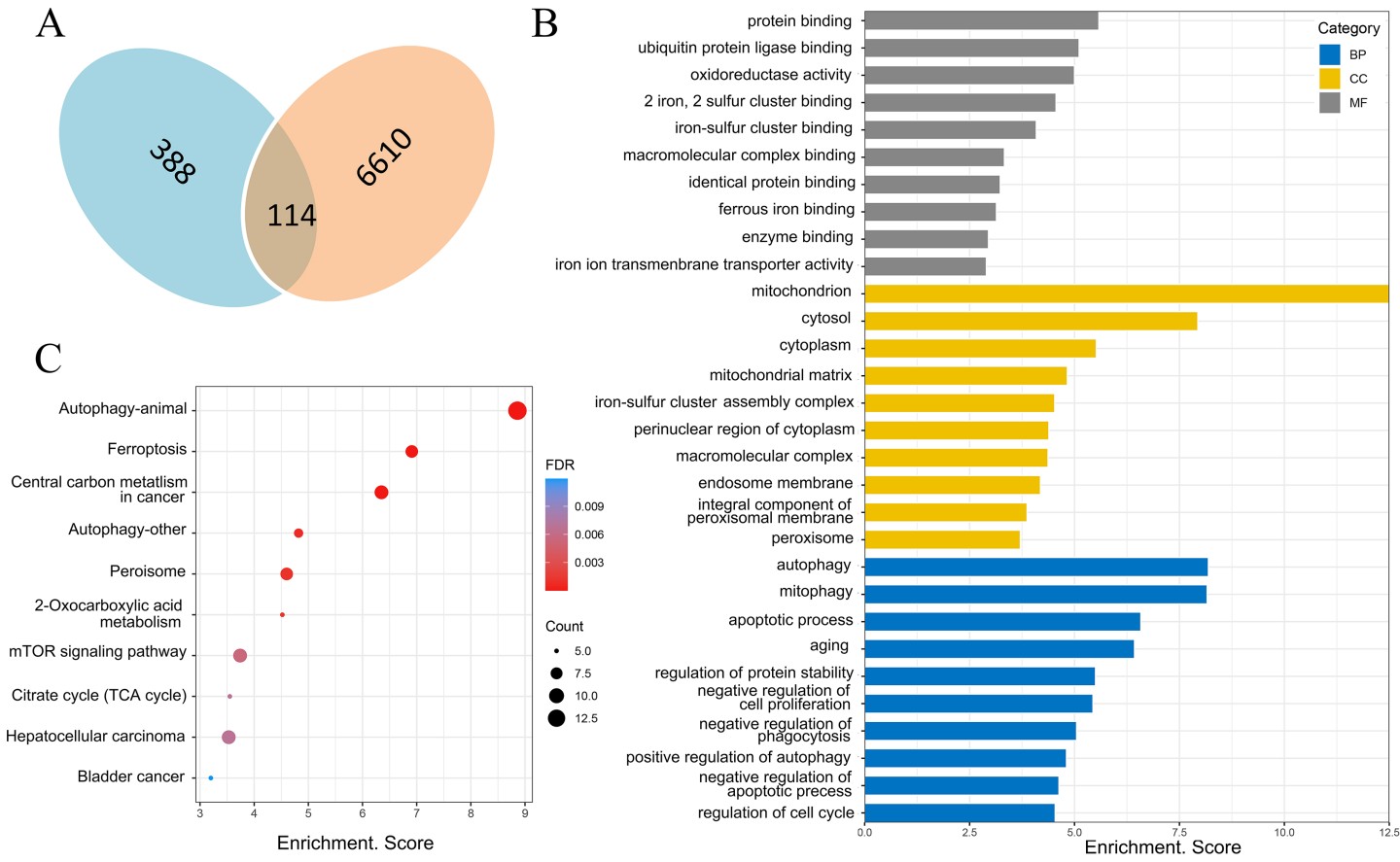

**Figure 6 Screening of ferroptosis-related genes and biological function analysis.** (A) Venn diagram, the blue areas represent ferroptosis-related genes in the FerrDb database, the orange areas represent the differentially expressed mRNAs. (B) GO analysis of ferroptosis-related genes in DOX-induced myocardial injury (DIMI), the top 10 GO terms from each category (BP, CC, MF) were listed. (C) KEGG pathway analysis of ferroptosis-related genes in DIMI, the top 10 pathways were listed.

positive regulation of autophagy (GO:0010508) in BP item, mitochondria (GO:0005739), iron-sulfur cluster assembly complex (GO:1990229), membrane (GO:0016020) in CC item, oxidoreductase activity (GO:0016491), two iron, two sulfur cluster binding (GO: 0051537), iron-sulfur cluster binding (GO:0051536) and iron ion transmembrane transporter activity (GO:0005381) in MF item were closely associated with ferroptosis. In KEGG pathway analysis results, 20 related signaling pathways were enriched, the top 10 pathways were shown in the form of bubble diagram in Fig. 6C, among which, ferroptosis was one of the important pathways. The above bioinformatics analysis results suggested that these 114 genes were involved in ferroptosis.

## PPA network was created with the STRING database and high score genes were scored by CytoHubba plugin through MCC algorithm

The PPA network of 114 ferroptosis-related genes in the DOX group was constructed through STRING database (Version 11.5), and score parameters was set to medium confidence (0.400) (Fig. 7A). PPA network data (Download date: 2022.9.23) was imported into Cytoscape software. Because CytoHubba plugin implements eleven node ranking
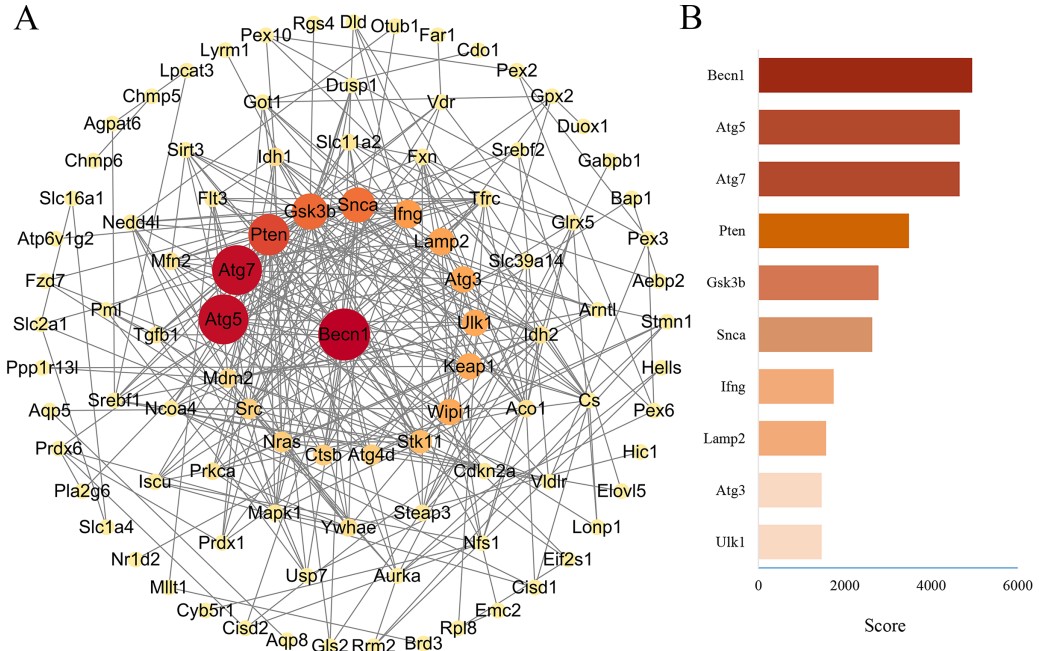

**Figure 7 PPA network of ferroptosis-related genes and the top 10 maximal clique centrality (MCC) score genes in DOX-induced myocardial injury (DIMI).** (A) PPA network of ferroptosis-related genes, the node color and size were set based on MCC score. (B) The top 10 MCC score genes scored using CytoHubba plugin, the high to low MCC score was represented by red and orange red bar.

methods to evaluate the importance of nodes in a biological network, MCC has a better performance on the precision of predicting important proteins, so the genes closely related to ferroptosis of PPA network were scored by the MCC algorithm through the CytoHubba plugin (*Chin et al., 2014*). Top ten genes were displayed in Fig. 7B, among them, the MCC score of Becn1 was the highest, and the microarray analysis results showed that Becn1 was the most significantly up-regulated gene among the 114 ferroptosis-related genes. Therefore, we speculated that Becn1 might play an important role in mediating ferroptosis involved in DIMI.

## Construction of ceRNA network of the highest MCC score gene Becn1

The miRNAs binding to Becn1 gene in DIMI were predicted using the Starbase database, and the prediction conditions were set as follows: predicted programs were miRanda and miRmap. The result showed that there were six miRNAs that could bind with Becn1. Then, the lncRNAs binding to each of the six miRNAs obtained above were predicted, the prediction conditions were set as follows: CLIP Data is medium stringency (≥2), and there were 19 lncRNAs binding to miRNA were obtained. Finally, these predicted lncRNAs were intersected with differentially expressed lncRNAs to obtain lncRNA/miRNA/mRNA ceRNA network. The results showed that the ceRNA network was composed of Becn1, three miRNAs and four lncRNAs, there are Rian/mmu-miR-145a-5p/Becn1, Tug1/mmu-miR-145a-5p/Becn1, Malat1/mmu-miR-30e-5p/Becn1 and H19/mmu-miR-299a-3p/

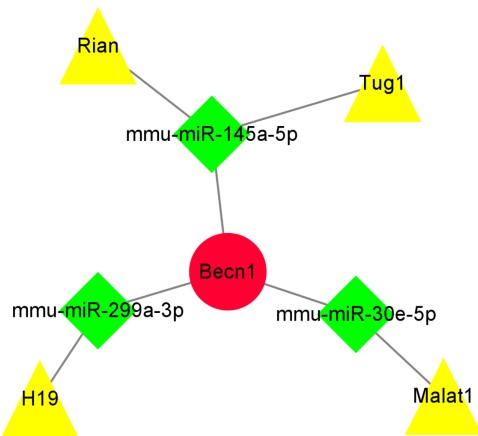

**Figure 8 The ceRNA network of the highest maximal clique centrality (MCC) score gene Becn1.** The ceRNA network was constructed based on predicted lncRNA-miRNA and miRNA-mRNA interactions. The ceRNA network include up-regulated lncRNAs, predicted miRNAs, and up-regulated mRNA. Red circle represents up-regulated mRNA Becn1, green diamond represents predicted miRNAs, yellow triangle represents up-regulated lncRNAs.

Becn1 axes respectively. The data of ceRNA network were imported into Cytoscape software for visual display (Fig. 8).

## The expressions of the major components were detected by qRT-PCR in ceRNA network

The qRT-PCR results showed that the expressions of Becn1 at mRNA level and four lncRNAs in the ceRNA network were significantly up-regulated in the DOX group compared with the CON group (Figs. 9A–9E), which had the same trend with microarray results (Fig. 9F). Hence, the expression levels of Becn1 mRNA and four lncRNAs met with the conditions to construct ceRNA network.

## DISCUSSION

It is well known that doxorubicin belongs to the first-line and broad-spectrum clinical antitumor drug, but due to its cardiotoxic side-effects, the incidence of heart failure in cancer patients is increased. Therefore, to explore the mechanisms of DOX induced myocardial injury and find the effective preventive measures can reduce the incidence of myocardial injury and improve the survival rate of cancer patients. Although numerous studies have elucidated the mechanisms underlying DIMI, the exact mechanism remains to do further research. Recent studies have found that ferroptosis is involved in DIMI (*Fang et al., 2019b*; *Tadokoro et al., 2020*), and alleviating ferroptosis may be a possible therapeutic strategy to prevent DIMI (*Kitakata et al., 2022*). Non-coding RNAs play an important role in DIMI (*Zhao et al., 2018*; *Hu et al., 2019*; *Lu et al., 2020*; *Zhan, Hu & Wang, 2020*), and the ceRNA regulation mechanism is one of the crucial ways (*Xia et al., 2020*), but few reports investigate the ceRNA network on regulating ferroptosis in DIMI. Therefore, this study aims to construct the lncRNA/miRNA/mRNA ceRNA network

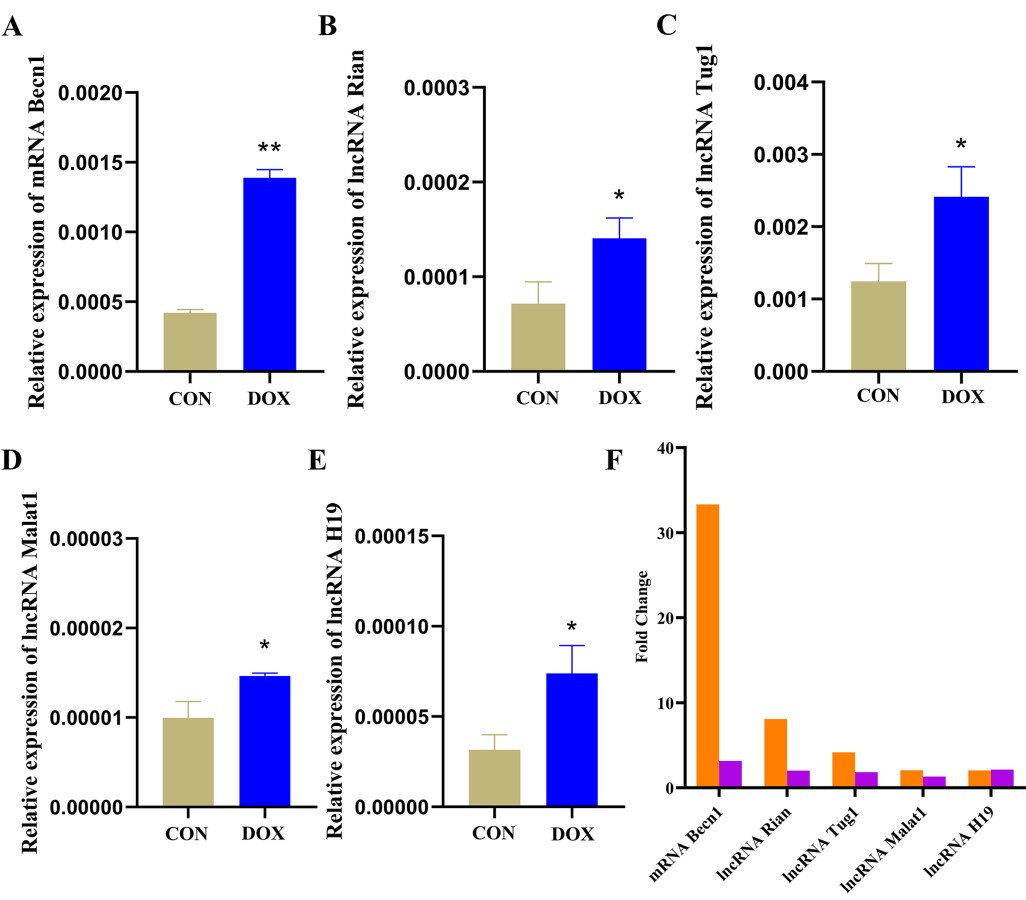

**Figure 9** **The expressions of the major components in ceRNA network.** (A–E) Relative expressions of the major mRNA and lncRNAs in ceRNA network. CON: Con group, marked with yellow bar; DOX: DOX group, marked with bule bar. (F) The expressions of the major mRNA and lncRNAs in ceRNA network compared to microarray results, microarray results were represented by orange bars and qRT-PCR results were represented by purple bars. $*P < 0.05$, $**P < 0.01$ *vs.* the CON group, the values are presented as mean ± SD ($n$ = 3).

through bioinformatics analysis technology in DIMI, it will provide the potential mechanisms and biomarkers for finding the effective therapeutic targets.

In this study, we used DOX to induce myocardial injury in mice model. The results showed that in comparison with the CON group, in the DOX group, the levels of serum LDH and CK-MB levels were significantly increased 3 days after DOX intervention. HE staining showed that the cardiomyocyte membranes and nucleus were abnormally stained with the destroyed structure. The changes of myocardial enzymes and histology suggested that DOX could cause myocardial injury. Meanwhile, we observed GPx4 protein expression in the DOX group was significantly lower than in the CON group, it suggested that ferroptosis was involved in DIMI, which was consistent with previous reports (*Tadokoro et al., 2020*; *Li et al., 2021*). Based on the above animal model, further research was carried out to identify the potential ceRNA mechanisms in DIMI.

In recent years, lncRNAs cause more and more attentions in disease development, recent investigations demonstrate that lncRNAs play a key role in cardiovascular disease,

which may be biomarkers or even the therapeutic targets for cardiovascular diseases (*Wang et al., 2021*). In this study, firstly, the differentially expressed lncRNAs and mRNAs were detected by microarray analysis in DIMI. The results showed that there were 10,265 differentially expressed lncRNAs (5,953 up-regulated and 4,312 down-regulated) and 6,610 differentially expressed mRNAs (2,136 up-regulated and 4,474 down-regulated) in the DOX group compared with the CON group. The microarray analysis results suggest that these differentially expressed lncRNAs and mRNAs may be involved in DIMI, and the possible mechanism requires further study.

Ferroptosis as a new programmed mode of cell death has been reported involved in DIMI (*Li et al., 2021*; *Kitakata et al., 2022*; *Chen et al., 2022*). Ferroptosis-related genes play a crucial role in regulation of ferroptosis in DIMI. So, we want to screen out ferroptosis-related genes from the differentially expressed mRNAs in DIMI. The 388 ferroptosis-related genes were obtained from the FerrDb database and intersected with the differentially expressed mRNAs, as a result, the 114 ferroptosis-related genes were screened. In order to further explore the potential biological functions of the 114 ferroptosis-related genes, GO analysis and KEGG pathway analysis were performed using the DAVID database. GO analysis results shows that the autophagy (GO:0006914) in BP item, mitochondria (GO:0005739), iron-sulfur cluster assembly complex (GO:1990229), membrane (GO:0016020) in CC item, oxidoreductase activity (GO:0016491), two iron, two sulfur cluster binding (GO:0051537), iron-sulfur cluster binding (GO:0051536) and iron ion transmembrane transporter activity (GO:0005381) in MF item were closely associated with ferroptosis. KEGG pathway analysis showed that ferroptosis was one of the important signaling pathways involved in DIMI. The above biological function analysis results suggested that ferroptosis was involved in DIMI, which was closely related to these 114 genes. Then the PPA network was obtained using the STRING database to analyze the interaction relationships between these 114 genes, and the ferroptosis-related genes were scored by the CytoHubba plugin using MCC algorithm in Cytoscape software. The result showed that Becn1 was the highest MCC score gene, which means that Becn1 was the most important protein in the PPA network. Combined with the microarray analysis results, it showed that Becn1 mRNA was highly expressed in the DOX group, and it was the most up-regulated gene among the 114 ferroptosis-related genes. Therefore, we speculated that Becn1 could play an important role in regulating ferroptosis in DIMI. Since previous studies reported that Becn1 was an important autophagy-related gene, but a recent finding (*Kang et al., 2018*) revealed that Becn1 was meanwhile a new driver of ferroptosis, which promoted ferroptosis through formatting Becn1-SLC7A11 complex to inhibit the cysteine and glutamate antiporter system $X_c^-$ activity in cancer cells. It also reported that Becn1 promoted hepatic stellate cells ferroptosis by suppressing xCT-driven Gpx4 expression (*Tan et al., 2022*), and Becn1 haploinsufficiency protected against low ambient temperature-induced myocardial remodeling and contractile dysfunction through inhibiting ferroptosis (*Yin et al., 2020*). Nonetheless, whether Becn1 participates in DIMI by promoting ferroptosis has not been reported. Therefore, we want to seek for the likely connection of Becn1 and ferroptosis in DIMI.

A prior study proposed a ceRNA hypothesis that mRNA, miRNA and lncRNA could crosstalk with each other to form a regulatory network (*Salmena et al., 2011*). In our study, 10,265 differentially expressed lncRNAs were detected by microarray analysis in the DOX group, accumulating evidences had suggested that lncRNAs could sponge miRNAs through ceRNA mechanism resulting in alterations in the miRNAs-regulated mRNA levels, so we further investigate the ceRNA mechanisms between these differentially expressed lncRNAs and Becn1 in regulating ferroptosis in DIMI. To systemically explore the potential ceRNA mechanisms between these differentially expressed lncRNAs and Becn1, miRNAs bound with Becn1 and lncRNAs bound with miRNAs were predicted respectively by Starbase database, then the predicted lncRNAs were intersected with the differentially up-regulated lncRNAs to obtain lncRNA/miRNA/mRNA ceRNA network, the results showed there were Rian/mmu-miR-145a-5p/Becn1, Tug1/mmu-miR-145a-5p/Becn1, Malat1/mmu-miR-30e-5p/Becn1 and H19/mmu-miR-299a-3p/Becn1 axes respectively. There have been reported that lncRNA Rian, lncRNA Tug1, lncRNA Malat1 and lncRNA H19 all serve as ceRNA involved in multiple biological process, such as apoptosis (*Wu et al., 2020*; *Yao et al., 2020*), ferroptosis (*Liang et al., 2022*; *Zhang et al., 2022*) and pyroptosis (*Sun, Mao & Ji, 2021*; *Kang et al., 2022*). Zhang and He reported that lncRNA Malat1/mmu-miR-30a/Becn1 was involved in myocardial infarction (*Zhang & He, 2020*). Furthermore, we also measured the expressions of Becn1 and the four lncRNAs (lncRNA Rian, Tug1, Malat1 and H19) by qRT-PCR, the results of Becn1 at mRNA expression and the four lncRNAs met the conditions to construct ceRNA network. All above further hinted that the ceRNA network we established has the theoretical credibility.

There are some limitations in the present study. First, only male mice were employed to observe the cardiotoxicity of DOX; it may be beneficial to select both female and male mice to explore the cardiotoxicity of DOX because DOX is a broad-spectrum antitumor chemotherapy drug in clinic. Second, we only selected single intraperitoneal injection of DOX to induce acute cardiotoxicity; it may did not truly reflect the dose-dependent cardiotoxicity of DOX in clinical chemotherapy. In the future, we will select both female and male mice and dose-dependent DOX to investigate the likely mechanisms of DIMI, and we will further to validate and investigate how Becn1 regulating ferroptosis and its regulatory role in ceRNA network in DIMI.

## CONCLUSIONS

In summary, we identified the differentially expressed lncRNAs and mRNAs in DIMI by microarray analysis, screened the highest MCC score gene Becn1 that regulated ferroptosis by bioinformatics analysis methods, and constructed the potential lncRNA/miRNA/mRNA ceRNA regulatory network. Our findings may be as a potential ceRNA mechanism in regulation of ferroptosis, and it can provide the objective for further research on the mechanism investigation of DIMI.

### Funding

This work was supported by the Anhui Province Education Key Project (grant no. KJ2021A0762), the 512 talent program of Bengbu Medical College (grant no. by51201102) and the key incubation project in Department of Basic Medicine, Bengbu Medical College (grant no. 2022JCYX02), China. The funders had no role in study design, data collection and analysis, decision to publish, or preparation of the manuscript.

### Grant Disclosures

The following grant information was disclosed by the authors:
Anhui Province Education Key Project: KJ2021A0762.
Bengbu Medical College: 51201102.
Department of Basic Medicine, Bengbu Medical College: 2022JCYX02.

### Competing Interests

The authors declare that they have no competing interests.

### Author Contributions

- Hongwei Ye conceived and designed the experiments, performed the experiments, analyzed the data, prepared figures and/or tables, authored or reviewed drafts of the article, and approved the final draft.
- Yuping Li conceived and designed the experiments, performed the experiments, analyzed the data, prepared figures and/or tables, and approved the final draft.
- Lu Li conceived and designed the experiments, performed the experiments, analyzed the data, prepared figures and/or tables, and approved the final draft.
- Yuhui Huang conceived and designed the experiments, performed the experiments, analyzed the data, prepared figures and/or tables, and approved the final draft.
- Jiahui Wang conceived and designed the experiments, performed the experiments, analyzed the data, prepared figures and/or tables, and approved the final draft.
- Qin Gao conceived and designed the experiments, analyzed the data, authored or reviewed drafts of the article, and approved the final draft.

### Animal Ethics

The following information was supplied relating to ethical approvals (*i.e.*, approving body and any reference numbers):

The Animal Management and Ethics Committee of Bengbu Medical College provided full approval for this research ([2022] 024).

### Microarray Data Deposition

The following information was supplied regarding the deposition of microarray data:

Data is available at the GEO database: GSE207737.

## Data Availability

The raw measurements are available in the Supplemental Files.

## Supplemental Information

Supplemental information for this article can be found online at http://dx.doi.org/10.7717/peerj.14767#supplemental-information.

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
