# Peer review of "Construction of a ceRNA network of regulated ferroptosis in doxorubicin-induced myocardial injury"

_PeerJ, doi:10.7717/peerj.14767_

## Round 0.1 · original submission · Major Revisions

Please check what reviewers suggested especially reviewer 2's suggestions.

Also a very important mistake: please check whether the differentially expressed genes calculation and enrichment analysis was done with false discovery rate correction. If not please apply an fdr correction otherwise the outcome is statistically not relevant.

Also please note that your work is in male mice, which is the standard for any side effect study, but DOX is used in breast cancer treatment. Please justify the use of only male mice and mention this limitation of the study.

Reviewer 1 ·

Basic reporting

The article is clear and unambiguous. Only few observations as should be revised the citation and references style (eg lines 57, 59, 60 ) throughout the text as well as the punctuation to be corrected (eg line 62)

Experimental design

The Experimental design is well established and represents the original primary research within Aims and Scope of the journal. Methods described with sufficient detail

Validity of the findings

The novelty is not clearly mentioned.

Additional comments

To make sure the citation and references style is in accordance with the journal requirements

·

Basic reporting

Ye et al carried out microarray detection and bioinformatic analysis to identify the ceRNA network affected by ferroptosis in doxorubicin-induced myocardial injury (DIMI) using a mouse model. The combination of experimental and computational approaches facilitated the identification of Becn1 and its ceRNA network that plays a potentially important role in ferroptosis in DIMI. The authors mostly collected the important articles in the field, they published their work mainly in an understandable way with accurate scientific language. The limitations of the study (e.g. gender bias due to using only male mice in the experiment) are not described in the manuscript. I would ask the authors to write it in the ‘Discussion’ section.
The figures and table are numbered well, however, there is no citation for Figures 2 and 3 in the text. The quality of them is adequate, but I missed a general figure about the workflow that could help the interpretation of the work. Another general problem is that at some points the figure description is too short (e.g. Figure 7) and does not give enough information to make the figures understandable with the figure description without the manuscript text.

Experimental design

The research question is well defined, looking for genes and their RNA regulatory network in ferroptosis is an essential step to identify the role of the process in DIMI. The introduction highlighted the dose-dependency of doxorubicin (DOX). Experiments examining the dose-dependency could increase the value of the study as well the gene/RNA expression profiling and the ceRNA network composition under different concentrations of DOX. These comments highlight potential future steps, the additional experiments are not expected to be carried out now. I would only ask the authors to write about this limitation in the ‘Discussion’ section.

Validity of the findings

Currently, the manuscript does not provide significant new information in terms of results, however, the workflow can identify other ferroptosis-related genes and their ceRNA network important in DIMI. Becn1 is a multifunctional protein with several interactors therefore it is not surprising that the network analysis highlighted it as a potential candidate with the highest MCC score for ferroptosis in DIMI. Also, the role of Becn1 is well-described in ferroptosis and autophagy and in myocardial injury as well. Moreover, its RNA regulatory network was analysed by Zhang and He (Zhang and He 2020), which article was not mentioned in the manuscript. Please read and cite the aforementioned article.
The authors mention in the discussion that Becn1 can form a complex with SLC7A11 hence promoting ferroptosis. The cited article also says that BECN1-dependent ferroptosis requires the formation of a BECN1-SLC7A11 complex. Hence, I suggest building up a ceRNA network using the mRNA for both of the genes and creating the network in control and DOX conditions separately.

Additional comments

There are a few grammar and spelling mistakes repeatedly occurring in the text: please correct ‘Microarray’ to ‘microarray’; ‘ferroptosis related’ to ‘ferroptosis-related’; ‘doxorubicin or DOX induced’ to ‘doxorubicin-induced’ / ‘DOX-induced’; String to STRING.
Also, a general correction for the manuscript is that space is needed between the text and the reference. Currently, this is missing in most cases. Finally, the manuscript contains many paragraphs consisting of one or two sentences. Please merge these sections or extend them.
I would like to add my further comments in the PDF document.

---

## Round 0.2 · Major Revisions

I am issuing a Major Revisions decision in response to your appeal of the recent rejection of this manuscript. in order to give you a chance to address the issues identified in the review process.

Please check further the previous comments of reviewer 2 and please incorporate them into the manuscript and not just in your response letter.

There is a statistical error is in the overrepresentation analysis. The authors used DAVID and they have not changed the background to the used gene chip. It should have been the used chip-specific background. Please add what background was used and why.

The STRING network that the authors used is NOT a protein-protein interaction network. It is a protein association network. The authors are using protein-protein interactions in the manuscript. Please add download date, versions, score parameters. It is required for proper scientific analysis.

Please describe the FDR correction method. Please describe in the manuscript.

Due to all the above-mentioned criticisms, the manuscript needs a thorough major revision.

· Appeal

Appeal

https://peerj.freshdesk.com/a/tickets/244179

Tue, 6 Sep 2022

> "We apologized for our misunderstanding of your suggestion. In our manuscript, about the data of differentially expressed genes calculation and enrichment analysis, the false discovery rate score were calculated, but were not corrected by false discovery rate.

Now, we recorrected the data with false discovery rate carefully, and compared the differences between P value correction and FDR correction in Table 1 (attachment). The results showed some changes in numbers of DelncRNAs, DemRNAs and ferroptosis-related genes, as well as numbers of enrichment analysis were emerged, but (). the results did not influence the key gene Becn1 and subsequent analyses. We consider the important part of the manuscript is not changed."

Sat, 24 Sep 2022

> "We have updated the manuscript according to all reviewers' comments and new analysis data based on FDR. Now, I send them to you by attachments, which include 4 documents: 1) Manuscript-tracked changes, 2) Manuscript-clean, 3) reply letter for reviewers' suggestions, and 4) answers to reviewer 2’s all comments."


· · Academic Editor

Reject

I have checked whether you used the fdr corrected genes for the analysis. You did not. Sadly, that is such a mistake, which basically invalidates the findings of the manuscript.

I wrote this at my evaluation at the first review round. Due to not changing I regretfully have to reject the manuscript.

---

## Round 0.3 · Minor Revisions

Please consider the reviewers comments.

·

Basic reporting

Ye et al submitted the edited manuscript in which the authors carried out microarray detection and bioinformatic analysis to identify the ceRNA network affected by ferroptosis in doxorubicin-induced myocardial injury (DIMI) using mouse model.

The authors corrected the text based on my suggestions: (1) including the limitation of the study in the Discussion section; (2) extension of short paragraphs; (3) re-numbering the figures and extending their description where it was needed; (4) they added a new workflow figure as I have requested previously, this has facilitated the understanding of the text.

However, there are a few minor things to correct: (1) mention the colour code in the Figure 4 description; (2) mention the value that has been used in Figure 5/A-B highlighted by purple or yellow, also use the same legend for the colours here (currently Fig5/A describes [-2, 2] values and Fig5/B describes [-1,1] values) and please improve its quality; (3) Figure 7/A is a STRING network including different edge and node colours, I appreciate that the authors attached a supplementary figure about the edge colour description, but still, this figure is unnecessarily complex and has a low resolution. Here, I would suggest to replace this with a Cytoscape network including these points and their interactions (without the detailed information regarding the interaction quality) and colour the nodes based on the MCC score; (4) the black text is not visible in the blue triangles on Figure 8, please change it; (5) mention that you used the driver, suppressor and marker categories from FerrDB in line 428.

Finally, please correct the following two misspellings in the text: (1) double space in line 447; (2) correct Benc1 to Becn1 in line 504

Experimental design

No comment.

Validity of the findings

The authors refused my suggestion to include SLC7A11 in the analysis because this mRNA didn't change significantly in the DOX-treated condition.

Although I think the current impact of their research is low, the workflow that the authors have built up is good, also regarding the future plans, I believe they can increase the validity of their findings.

---

## Round 0.4 · Minor Revisions

Dear Authors, I checked the paper and minor editorial comments are left.

Please change the PPI network everywhere in the manuscript protein-protein association networks, what you have used. This includes on Figure 1.
Otherwise the manuscript is ready for publication.

---

## Round 0.5 · accepted · Accept

Thank you. Merry Christmas and Happy New Year!